# Planning and Execution in Multi-Agent Path Finding: Models and Algorithms

**Primary Keywords:** *Multi-Agent Planning*

## Abstract

In applications of Multi-Agent Path Finding (MAPF), it is often the sum of planning and execution times that needs to be minimised (i.e., the *Goal Achievement Time*). Yet current methods seldom optimise for this objective. Optimal algorithms reduce execution time, but may require exponential planning time. Non-optimal algorithms reduce planning time, but at the expense of increased path length. To address these limitations we introduce PIE (Planning and Improving while Executing), a new framework for concurrent planning and execution in MAPF. We show how different instantiations of PIE affect practical performance, including initial planning time, action commitment time and concurrent vs. sequential planning and execution. We then adapt PIE to Lifelong MAPF, a popular application setting where agents are continuously assigned new goals and where additional decisions are required to ensure feasibility. We examine a variety of different approaches to overcome these challenges and we conduct comparative experiments vs. recently proposed alternatives. Results show that PIE substantially outperforms existing methods for One-shot and Lifelong MAPF.

## Introduction

Multi-Agent Path Finding (MAPF) (Stern et al. 2019) is the problem of finding collision-free paths for a team of moving agents. Efficiently solving MAPF is crucial for many real-world applications, such as automated warehouses (Wurman, D'Andrea, and Mountz 2008), automated intersections (Li et al. 2023) and computer games (Silver 2005).

When solving MAPF problems existing studies typically assume that necessary computation time is available up front (Lam et al. 2022; Li et al. 2021c,a; Okumura 2023). Smaller times are preferable but typically not reflected in the corresponding objective functions, which instead aim to minimise action costs; e.g, Makespan (Yu and LaValle 2013) or Sum-of-Costs (Stern et al. 2019). The main advantage of this approach, sometimes known as *offline planning* is that execution times are as small as possible, subject to time-out limits (which can range from seconds to hours). The main drawback to offline planning is a mismatch between the model assumptions and the requirements of real applications, which can be entirely *online*. In other words, if a plan is not immediately available real-world agents simply *wait in place*, until the planner can provide instructions.

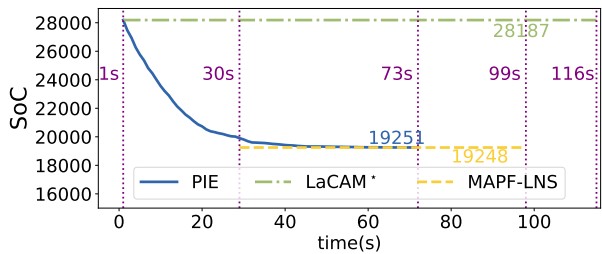

Figure 1: Planning and execution costs for 3 different MAPF algorithms on a small grid map; `random-32-32-20` with 400 agents and unit action costs. PIE and LaCAM* compute the same initial solution and begin execution after 1s. MAPF-LNS plans for a further 29s then begins execution.

An alternative approach, which can reduce up-front delays, involves decomposing a MAPF problem into a sequence of smaller sub-problems; e.g., each having a limited time horizon (Švancara et al. 2019; Li et al. 2021d; Morag, Stern, and Felner 2023). The resulting *interleaved planning* model has the advantage that agents are provided with instructions sooner. The main drawback is an increase in execution costs, as less time is available for each planning episode. Another drawback is that the planner runs until a solution is found, which may take more or less time, depending on the sub-problem at hand. In other words, planning episodes do not necessarily have a fixed duration.

In this work we propose a new *concurrent planning* framework which we call PIE: Planning and Improving while Executing. PIE leverages fast solvers to quickly compute and commit to a small number of actions for each agent. During execution of these actions, PIE optimises the remaining paths of agents and then commits to a new set of actions. Concurrent algorithms have been previously studied in single-agent search (Korf 1990) where they are known to reduce waiting time for agents and improve *Goal Achievement Time* (GAT) (Gu et al. 2022).

As a motivating example consider Figure 1, where we illustrate the concrete advantages of PIE (in blue) compared with two leading offline planners: MAPF-LNS (Li et al. 2021a) (the best known algorithm for anytime MAPF, in yellow) and LaCAM* (Okumura 2023) (the best known algo-

rithm for scalable MAPF, in green). The graph shows Sum-of-(executed action)-Cost (SoC) over time, with the end-point of each line indicating the end of execution (i.e., GAT for the last arriving agent). We make three observations: (i) PIE finishes executing substantially faster than either offline planner; (ii) the execution costs for PIE are very similar to MAPF-LNS, which requires 29 seconds of additional compute; (iii) advantages are magnified when considering up-front wait costs: +400 for PIE and LaCAM$^\star$ and +12000 (400×30) for MAPF-LNS.

We describe the general PIE framework and the main decision variables required by the model. We then analyse and experiment with two distinct instantiations: PIE for one-shot MAPF (where each agent has a single target) and PIE for Lifelong MAPF (where agents are continuously assigned new tasks). Results show substantial improvements for Goal Achievement Time for one-shot MAPF iand substantially higher thoughput for Lifelong MAPF, in comparison to leading methods from the area.

## Existing Models and Algorithms

**Offline Algorithms for Multi Agent Path Finding:** *Lazy Constraint Addition Search for MAPF* (LaCAM$^\star$) (Okumura 2023) is a *highly scalable* suboptimal algorithm for one-shot MAPF. LaCAM$^\star$ is a two-level search. In the high-level, the algorithm explores configurations of agents, i.e., a sequence of non-repeated vertices, one for each agent, in a depth-first search style. Each high-level node is associated with a configuration and a constraint tree. The configuration, which satisfies the associate constraint tree, is efficiently generated by Priority Inheritance with Backtracking (PIBT), a rule-based MAPF solver (Okumura et al. 2019). If the configuration generation fails, LaCAM$^\star$ does not immediately discard the node. Instead, it gradually grows the constraint tree with the low-level search that governs agent movements in subsequent configurations. LaCAM$^\star$ further improves its performance by discarding duplicate configurations and enhancing PIBT with a pattern-based swap operation (Luna and Bekris 2011; De Wilde, Ter Mors, and Witteveen 2014).

*Large Neighbourhood Search for MAPF* (MAPF-LNS) (Li et al. 2021a) is the current state-of-the-art anytime search algorithm for one-shot MAPF. It aims to improve MAPF solutions to near-optimal within a given time budget by replanning subsets of agents using *Large Neighbourhood Search*. MAPF-LNS initiates with an initial solution and iteratively modifies it by selecting a subset of agents as the *neighbourhood*. This *neighbourhood* is heuristically chosen by three strategies (agent-based, map-based and random), and the associated paths are removed and replanned to ensure collision avoidance. If successful, the new paths replace the existing ones, contributing to a better solution. The neighbourhood selection strategy is *adapted* during execution to use neighbourhoods that have more recently improved paths, and a tabu-list is used to avoid continually examining the same sets of agents. The low-level search for getting the initial solution and replanning in LNS uses *Prioritised Planning* (PP) (Silver 2005) PP simply plans for agents in an order, e.g., randomly sampled. It plans paths for each agent while

treating other higher-priority agents' paths (including those not being replanned) as temporal obstacles.

*MAPF-LNS2* (Li et al. 2022) is a sub-optimal MAPF algorithm based on large neighbourhood search. It starts by calling PP to find paths for a MAPF instance. For the agents that fail to obtain a conflict-free path, MAPF-LNS2 plans paths for them while minimising the number of collisions using Safe Interval Path Planning with Soft Constraints (SIPPS) (Li et al. 2022; Phillips and Likhachev 2011). Then MAPF-LNS2 selects the neighbourhood agents based on the heuristic, destroys and repairs the neighbourhood paths to reduce the total number of conflicts in the solution. This process iterates until the current repaired solution is collision-free.

**Interleaved Planning and Execution in Multi Agent Path Finding:** One similar idea is called Rolling-Horizon Collision Resolution (RHCR) (Li et al. 2021d), which is designed for Lifelong MAPF. In the RHCR framework, the solvers only need to plan for a MAPF solution that is collision-free within a window. By doing so, agents can quickly start executing the windowed solution. RHCR runs this planning and execution of the partial solutions sequentially. It performs well in large-scale warehouses against other Lifelong MAPF algorithms in terms of the number of goals completed, but the runtime of the planning algorithm is regarded as free of cost in the simulation. In other words, the actual GAT needs to be increased by adding the waiting time for planning. In (Morag, Stern, and Felner 2023), authors also applies RHCR in their Lifelong MAPF framework. In their experiments, they set the planning time limit to be the same as the execution time per window, which can be adapted to our problem settings (planning while executing). However, the low-level planner in RHCR cannot guarantee to compute the windowed solutions within the strict 'execution move' time limit, it suffers from a high failure rate when scaling up with more agents. In addition, once the solver finds a solution, it will terminate. Thus, there is no chance to improve the returned solutions.

**Concurrent Path Finding Algorithms:** Concurrent planning and execution are mostly studied in single agent planning, called Real-time Heuristic Search (RTHS) (Korf 1990). During the execution, RTHS algorithms perform a constant amount of expansions (or within a fixed time limit) as look-ahead in the search tree. Then the algorithm commits the next $k$ actions and re-roots the tree from the last location of the committed actions. It can be regarded as planning while executing when the look-ahead time limit is the same as the time on execution. Since RTHS generates only partial solutions during each commit, studies in this direction focus on improving the look-ahead overheads and developing dynamic commitment strategies to optimise the feasibility of the final plan (Gu et al. 2022; Elboher et al. 2023; Koenig and Sun 2009).

In (Sigurdson et al. 2018), authors consider this real-time setting for MAPF, they run individual RTHS for each agent and avoid collisions for agents nearby, i.e., within a given "vision" limit. Since RTHS search performs limited look-ahead, the commitment has a risk of making incorrect action choices, such as an action leading to planning failure. Furthermore, the complexity of MAPF environments makes

the problem harder to solve in a single search tree, i.e., collision avoidance between agents. As a result, RTHS often suffers from a low success rate and agents may reach dead-ends when pushed by others.

## Problem Setup

**MAPF:** The input is an undirected 4-connected gridmap $G = (V, E)$, and a set of $m$ agents $A = \{a_1...a_m\}$. Each agent has a start location $s_i \in V$ and a goal location $g_i \in V$. Time is assumed to be discrete. At each timestep, each agent takes one action: either *wait* at the current location, or *move* to an adjacent location. A path is a sequence of actions that can transit an agent from its start to goal location. The length (or cost) of a path is its number of actions. A plan is a set of paths, one for each agent. A conflict occurs in a plan if two agents would occupy the same vertex at the same timestep, or if they would pass through the same edge in opposite directions at the same timestep. A MAPF solution $\pi$ is a conflict-free plan.

**One-shot and Lifelong:** Conventional MAPF focuses on solving the "one-shot" version of this problem, which is solved when all agents are at their goal. By comparison, Lifelong MAPF is a MAPF variant in which an agent receives a new goal once it reaches its current goal (Li et al. 2021d; Morag, Stern, and Felner 2023). This process continues until a simulation time horizon $T$ is reached. The goals are assigned by the Task Oracle (TO), which can reveal a number of subsequent goals to each agent. In this work, we reveal one goal at a time.

**Concurrent Planning and Execution:** For concurrent planning and execution in MAPF, Agents can execute with a partial solution. That is, a MAPF solver can commit only $k$ steps of actions in the path to the agents, denoted as $\pi_k$. The committed path $\pi_k$ is now locked and cannot be changed. During the execution of $\pi_k$, the solver can further plan for the uncommitted path towards the goal or do nothing. After the $k$ timesteps execution, agents will wait until receiving the next partial solution to execute. This process continues until the problem is solved, i.e., all agents reach their goal locations. We make additional simplifying assumptions that:

- Both the execution time of each action and planning time is counted as an integer value of seconds;
- The execution is perfect with no delay and the communication time for committed actions is free of cost.

**Objective Functions:** Normally in conventional MAPF, the objective is to minimise the Sum of path Costs (SoC), which is the sum of the execution time for each agent. In concurrent planning and execution, both planning time and execution time are measured together. In single agent planning, this is called Goal Achievement Time (GAT) (Gu et al. 2022), which is the time for a single agent from the planning start to reach the goal location in execution. For MAPF, the GAT is defined for each agent, as the sum of planning time and path length. We simultaneously optimise GAT for all agents, which is measured as the Sum of the Goal Achievement Times (SGAT).

For Lifelong MAPF, there is no fixed goal. Instead, the objective is to maximise the throughput, i.e., the average

---

**Algorithm 1: PIE Framework**

**Input:** $\langle G, A \rangle$; $T_{init}$, initial solution planning time limit; $T_{action}$, execution time for one action; $k$, number of actions per commit.
1: $T_{exec} \leftarrow T_{action} * k$
2: $\pi \leftarrow \text{Plan\_Improve}(\langle G, A \rangle, \emptyset, T_{init})$
3: Commit $\pi_k$
4: $\pi \leftarrow \pi \setminus \pi_k$
5: **while** (Execution of $\pi_k$) **do**
6:     Update $A.starts$, $A.goals$ from TO
7:     $\pi \leftarrow \text{Plan\_Improve}(\langle G, A \rangle, \pi, T\_exec)$
8:     Commit $\pi_k$
9:     $\pi \leftarrow \pi \setminus \pi_k$

---

number of goals reached per timestep by time $T$ (equivalently, the throughput can be understood as maximising the number of goals reached).

## Planning and Improving while Executing

In this section, we describe a new concurrent planning and execution framework for MAPF. Our model has several components, which must be instantiated:

- Initial Planning Time ($T_{init}$): this variable is the time allowed to compute an initial solution. $T_{init}$ is also counted as a waiting cost for every agent in the SGAT

- How Long to Commit ($k$): this variable is the number of actions that the agents commit to during each execution phase. Once committed, these $k$ actions cannot be changed.

- Execution time ($T_{action}$): this variable specifies the time required to execute a single action. Multiplying by $k$ gives the time available for the planner to compute the next set of actions before the agents incur additional waiting time.

- Planner: the main ingredient in PIE is the planner. We suggest algorithms that can incrementally improve the solution until time out. However, any MAPF planner can be used.

Pseudocode for the PIE framework is shown in Algorithm 1, we take as input $T_{init}$, $k$, $T_{action}$, the map $G$ and a set of agents $A$ with initially assigned start and goal locations. PIE starts by generating an initial solution $\pi$ and improves $\pi$ within the runtime limit $T_{init}$ (line 2). The algorithm then commits the first $k$ actions of $\pi$, and updates the solution $\pi$ to be the uncommitted part of the solution. Then agents iteratively commit and execute (lines 5-9). The loop terminates when all agents stay at goals. During each execution, the planner will plan and improve the uncommitted part of the solution with runtime limit $T_{exec}$ (line 7). After planning, the planner commits the next $k$ actions and updates the uncommitted solution $\pi$ (line 8-9).

## PIE for One-Shot MAPF

In this section, we show how to instantiate PIE for one-shot MAPF. The approach combines LaCAM$^\star$, which we used to compute fast feasible solutions, and MAPF-LNS, which we use to improve the costs of uncommitted actions.

---

**Algorithm 2: Plan_Improve for MAPF**

**Input:** $\langle G, A \rangle; \pi; T\_max$
1: **if** $\pi$ is a partial solution or $\pi$ is $\emptyset$ **then**
2:     $\pi \leftarrow \text{LaCAM}^\star(\langle G, A \rangle, \pi, T\_max)$
3: $T_{remain} = T\_max-$ runtime of LaCAM$^\star$
4: $\pi \leftarrow \text{MAPF-LNS}(\langle G, A \rangle, \pi, T_{remain})$
5: **Return** $\pi$

---

MAPF-LNS utilises MAPF-LNS2 to get initial solutions, but MAPF-LNS2 can be ineffective when given only a short time limit but a large number of agents (Shen et al. 2023) i.e., time of executing one action. Therefore, we use LaCAM$^\star$ instead, which outperforms MAPF-LNS2 in scalability. We then modify LaCAM$^\star$ to return partial solutions on time-out, in which we select the best node explored so far as the partial solution. The best node is measured by the number of goals reached with tie-breaking on the maximum depth of the search node to ensure as many steps possible in the solution do not have conflicts.

Pseudo-code is shown in Algorithm 2. First, we generate an initial solution if there is no current solution (line 1-2), and use the remaining time to run MAPF-LNS to improve the solution (line 3-4). Noticing that if the uncommitted $\pi$ is a partial solution or not feasible, we discard the plan and compute anew. [1] When improving (line 4), we maintain the adaptive weights for MAPF-LNS, which affect the choice of destroy heuristics, and tabu list, which prevents MAPF-LNS from keep selecting the same group of agents for replanning.

## PIE for Lifelong MAPF

Lifelong MAPF is more time-sensitive because real-world applications like automated warehouses require consistent and real-time operation. In Lifelong settings, neither LaCAM$^\star$ nor MAPF-LNS can be directly applied due to the following reasons:

- **Frequent Replan**: Agents constantly receive new goals during ongoing execution, and new conflict-free paths to new goals are constantly required.

- **After-Goal Decision**: In Lifelong settings, the planner is not aware of where the agent should go after reaching the goal, because the new goals are only revealed by TO after the agent reaches its goal on the map.

- **Failure Path Finding with Same Goal**: In Lifelong, more than one agent may have the same goal location. Applying MAPF planners to such problems will lead to search failure.

Algorithm 3 shows the overview of Plan_Improve adapted to Lifelong MAPF, which addresses the above challenges. Comparing with Algorithm 2, Algorithm 3 have additional lines (line 3-11) to replan paths to new goals.

We will discuss different after-goal decision strategies, planner decisions and their trade-offs in the following sections.

---

[1]As in experiments, we observe that LaCAM$^\star$ mostly succeeds within seconds. Therefore we simply restart LaCAM$^\star$ in the next iteration if it fails to get an initial solution.

---

**Algorithm 3: Plan_Improve for Lifelong MAPF**

**Input:** $\langle G, A \rangle; \pi; T\_max$.
1: **if** $\pi$ is a partial solution or $\pi$ is $\emptyset$ **then**
2:     $\pi \leftarrow \text{LaCAM}^\star(\langle G, A \rangle, \pi, T\_max)$
3: **else**
4:     $A^{'} \leftarrow$ agents that have a new goal
5:     **if** $A^{'}$ is not $\emptyset$ **then**
6:         **if** Replan strategy is Replan All **then**
7:             $\pi \leftarrow \text{LaCAM}^\star(\langle G, A \rangle, \pi, T\_max)$
8:         **if** Replan strategy is Replan Affected **then**
9:             run PP for $A^{'}$
10:            **if** No solution **then**
11:                $\pi \leftarrow \text{MAPF-LNS2}(\langle G, A^{'} \rangle, \pi, T\_max)$
12: $T_{remain} = T\_max-$ runtime of line 1-11
13: $\pi \leftarrow \text{MAPF-LNS}(\langle G, A \rangle, \pi, T_{remain})$
14: **if** $\pi$ has a collision in current commit window **then**
15:     Failure resolution for $\pi$
16: **Return** $\pi$

---

## Replan Strageties

Different approaches to replanning are discussed in the literature, including replanning all agents, replanning a single agent and replanning a single group (Švancara et al. 2019). Replanning a single agent in Lifelong MAPF may find the agent has no path if all other agents' paths are locked, so it is not applicable to our problem. We consider the other two approaches, Replan All and Replan Affected.

**Replan All.** The simplest approach is to replan for all the agents once there are new goals. For replan all, once there are new goals (line 3-5), we simply use LaCAM$^\star$ to plan paths for all agents from their current positions and replace $\pi$ with the new solution (line 6-11). Replan All with LaCAM$^\star$ is fast and with high scalability. However, such an approach wastes the previous effort on path improvement, as the uncommitted MAPF-LNS improved paths of agents without new goals are completely removed and replaced by LaCAM$^\star$ solution, leading to lower throughput.

**Replan Affected.** To maintain the search efforts in previous MAPF-LNS improvements, we minimise the agents that need to be replanned by using MAPF-LNS2 (line 8-11). We first use PP to plan for only the agents that have new goals while regarding other agents as dynamic obstacles (line 9). If PP fails on some agents, we then enable MAPF-LNS2 to repair the incomplete solution. (line 11). Replan Affected preserves the existing paths as much as possible by replanning for a small group of agents. It helps maintain the solution quality, but PP and MAPF-LNS2 may fail to find conflict-free solutions in a short time limit, as the underlying time-space search is slow in congested situations. Thus, a failure resolution policy will be introduced later to handle the situation.

## After-Goal Decision in Planner

Many one-shot MAPF planners, like PP and MAPF-LNS(2), assume agents stay at goal forever after reaching the goal. If we do nothing when planning for lifelong MAPF with multiple agents having the same goal, planners will fail to find a solution as the goal state includes conflict. If we assume the

agent disappears after reaching the goal, we may commit a conflict solution to the simulator, as agents never disappear in the execution simulator. In this section, we discuss three after-goal decisions for the planner, to ensure collision-free actions for commitment.

**Disappear Immediately.** In (Švancara et al. 2019), authors treat agents as disappearing immediately after reaching goals. We apply the same idea as one choice. However, if the commit window is more than one step, disappearing immediately may cause problems since we commit actions that assume agents disappear, but they do not actually disappear in the simulator thus causing conflicts.

**Stay at Dummy Goal.** In (Li et al. 2021d), authors assign a final dummy goal to each agent where it can safely stay if an agent has no goal. In their problem setting, the map has some areas reserved for dummy goals, e.g. robot charging stations, which are never selected as real goals and are unlikely to block other agents' paths. However, not all maps have such a setup. Therefore, we design a dummy goal selection rule based on the *degree* of each location, which indicates how many traversable locations a location connects to. First, we collect all vertices with the highest degree as candidate dummy goals. If the number of candidates is not enough (less than the number of agents), we continue the same collection process that adds the set of vertices with one less degree as candidates. This process continues until we have at least as many candidates as agents, and then we randomly select dummy goals for each agent from the candidates.

Our dummy goal selection rule tries to minimise the chance that the selected dummy goal is a must traverse through location for other agents Our planner is modified to plan paths from starts to current goals, and then to dummy goals. However, such an approach wastes efforts to plan paths that will be thrown away once the agent reaches the current goal and gets a new goal, resulting in longer planning runtime.

**Disappear After Commit Window.** To avoid the wasted planning efforts, we extend the Dummy Goal approach. The new approach moves agents towards their dummy goal when reaching their current goals, but the paths are only planned up to the end of the commit window. This agent is then regarded as disappearing after the commit window. When committing actions including reaching goal events, these agents immediately get new goals for future windows, thus the approach never commits actions with agents "disappear".

### Failure Resolution Policy Based on MCP

Both Replan Affected and Disappear Immediately may return a conflicting solution in the upcoming commit window. More importantly, the planner may return a collision solution if the time limit is tight. Thus, we design a failure resolution policy based on the Minimal Communication Policy (MCP) (Ma, Kumar, and Koenig 2017). MCP is a robust execution policy that aims to handle unexpected delays during plan execution. In (Li et al. 2021b), authors extended the usage of MCP, they rebuilt a complete MAPF plan by simulating the execution using MCP until all agents reached their

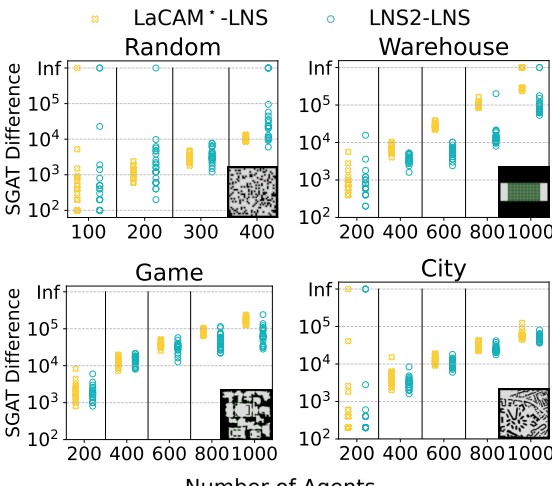

Figure 2: SGAT difference between "optimised to PIE SoC before executing" and PIE. "Inf" means the offline LNS solver times out when computing a solution or optimising a solution to reach the PIE SoC.

targets if a delay happened.

In this work, we use a modified MCP to rebuild a plan that removes any collision within the commit window. First, we record the visiting events order for each vertex according to the committed plan $\pi_k$, where each visiting event is a set that records the agent that will visit the vertex at each timestep (we omit timesteps with no visits). When building the visiting events order, multiple agents can visit at the same time, indicating vertex collisions in the current plan. For example, a vertex $v$ has visiting even order $o = [\{a_1\}, \{a_2, a_3\}, \{a_2, a_4\}, \{a_5\}]$ indicating $a_1$ visits $v$ first, $a_2$ and $a_3$ then visit $v$ at the same time, $a_2$ and $a_4$ collide at $v$ as $a_2$ waits at $v$, and $a_5$ visits $v$ after $a_2$ and $a_4$ leaves.

We start with all agents in their current state recorded on the paths. At each timestep, the simulation increments an agent, which is not marked as "done" yet, to the next state recorded in its path if and only if *Condition 1:* the agent is included in the top visiting event of its current vertex, and *Condition 2:* it is also included in the top visiting event for the target vertex, are satisfied. The simulation erases an agent from the top visiting event of its current vertex when its state increments to the next state and pops the event if the event is empty. Otherwise, the simulation issues a stay at the current state for this timestep.

If an agent reaches the last state on its path, it is marked as "done" and erased from the corresponding visiting event. The rebuild finishes when all agents are marked as "done". When rebuilding paths, this *generalised* MCP pauses agents competing for the same vertex at the same time within the committed window and allows conflict actions outside the committed window. So that, the committed actions are conflict-free and paths outside the committed window are given to the planner for optimisation while executing.

**The Necessity of Condition 1.** State increment condition

1 is necessary to guarantee the correct execution to rebuild the collision-delayed plan, with the original MCP only having rules similar to condition 2.

Assuming, agent $a_2$ is with path $p_2 = [v^-, v, v, v^+]$. When building the visiting event order, the original MCP may ignore the wait on $v$ and only consider which agent visits $v$ before $a_2$ and which agent is after $a_2$. But here, the MCP visiting event order is built upon a collision plan. To distinguish between the situation that $a_2$ is in collision with both $a_3$ and $a_4$ and $a_3$ visit the $v$ earlier than $a_4$, we always record each visiting (occupation) event caused by the wait in the visiting event order.

In this case, when $a_1$ left $v$ then $a_2$ enters $v$ the current order or $v$ is $o = [\{a_2, a_3\}, \{a_2, a_4\}, \{a_5\}]$ and the current state of $a_2$ progressed to $p_2[1] = v$. $a_2$ then is allowed to progress to $p_2[2] = v$ as the current vertex and next vertex are both $v$ and $a_2 \in o[0]$. Then the visiting order is updated to $o = [\{a_3\}, \{a_2, a_4\}, \{a_5\}]$. Assuming $a_3$ is delayed somewhere else and will not visit $v$ in a few timesteps. If condition 1 does not exist, $a_2$ will increment to $p_2[3] = v^+$ but as $a_2 \notin o[0]$, the pop $a_2$ from the top event of $o$ operation will miss $a_2$ and leaves its occupation in $o$ forever. Thus we either need extra mechanisms to pop from the correct event, or we simply use condition 1 to maintain the visiting order correctly. We choose to use condition 1, where $a_2$ only increments to $p_2[3]$ after $a_3$ clears its visiting event.

# Experiments

We implement PIE in C++[2] and conduct experiments in a Nectar Cloud VM instance with 16GB RAM, 8 AMD EPYC-Rome CPUs for one-shot MAPF problems and another Nectar Cloud VM instance with 32GB RAM, 16 AMD EPYC-Rome CPUs for Lifelong MAPF problems.

## One-shot MAPF

We conduct experiments on four different maps using grid-based Multi-Agent Path Finding (MAPF) benchmarks

---
[2]Code is at https://publish_after_acceptance

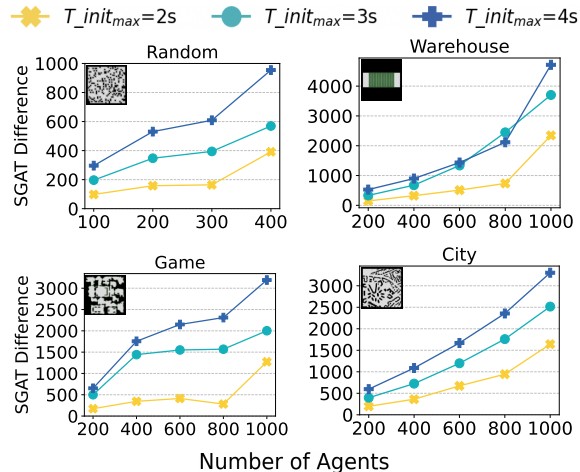

Figure 4: Average SGAT Difference between initial planning time is 2,3,4s and initial planning time is 1s.

sourced from (Stern et al. 2019) spanning different domains. These maps are named *random-32-32-10* (referred to as Random), *warehouse-10-20-10-2-1* (referred to as Warehouse), *ht_mansion_n* (referred to as Game), and *Paris_1_256* (referred to as City). For each map, we test all 25 random scenario files available in the benchmark sets. We vary the number of agents up to the maximum number of agents from the benchmark sets, which are from 100 to 400, increasing by 100 for Random, and from 200 to 1000, increasing by 200 for other maps. The runtime limit for the initial solution runtime limit ($T\_init$) is set to 1 second, which is the best setup we found from experiments.

**Experiment 1: Optimised Before Execution.** To show the necessity of optimising while execution and the price of optimising the solution while agents waiting at starts. We evaluate PIE with commit = 1 as a baseline and record the corresponding $SoC$ for each instance. We then simulate an optimising before execution situation by running an offline MAPF-LNS to get an equal or better SoC and

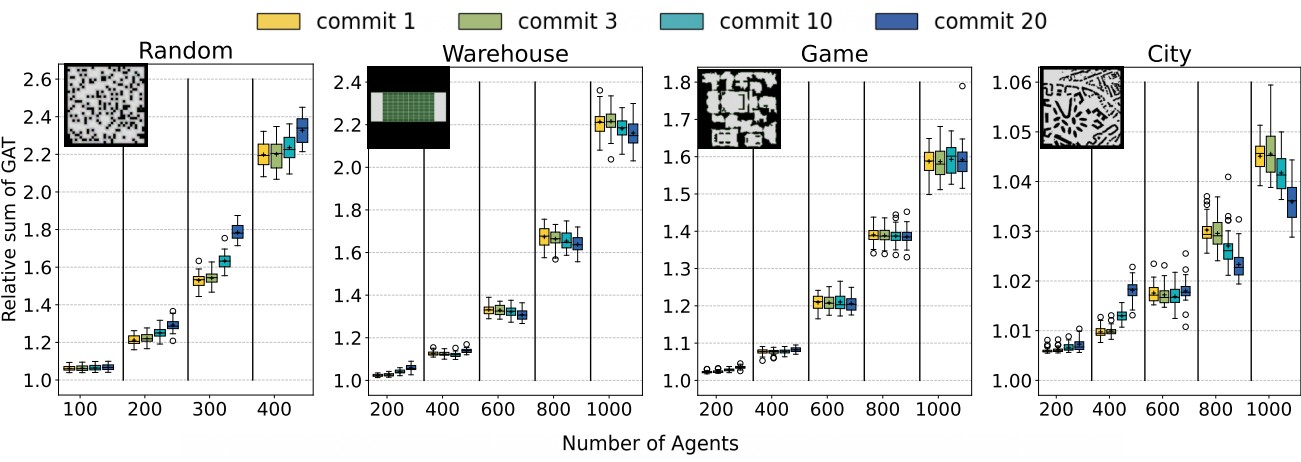

Figure 3: Relative SGAT (SGAT divided by lower bound) with different commit windows 1, 3, 10, 20.

evaluate its SGAT. That is, we terminate MAPF-LNS when $SoC_{current} \leq SoC_{PIE}$. We evaluated MAPF-LNS with both MAPF-LNS2 and LaCAM$^\star$ as the initial solution solver. The runtime limit is set to 300 seconds. Figure 2 shows the SGAT difference between "optimise first to get a good solution before execution" and "immediately starting with PIE". This is calculated by the SGAT of running offline MAPF-LNS to get an equal or better solution minus the SGAT of PIE. The cost of optimising offline first grows substantially when scaling, because the runtime for computing an equal or better solution grows, causing substantial delays when agents are waiting in place. It is worth noting that although MAPF-LNS2 as an initial solver outperforms LaCAM$^\star$ in Warehouse and Game maps offline, we observe that MAPF-LNS2 takes the majority of the runtime for computing an initial collision-free solution, while LaCAM$^\star$ always finishes in less than 1s. Therefore for PIE, we use LaCAM$^\star$ as the initial solver to start executing as quickly as possible.

**Experiment 2: Initial Planning Time.** We set commit = 1 as a baseline, and evaluate for PIE, to see if using a longer initial planning time ($T_{init}$) before starting yields better solutions. Figure 4 shows the difference in average SGAT of $T_{init} = 2, 3, 4$s compared to $T_{init} = 1$s. The difference is calculated by the SGAT of $T_{init} = n$ minus the SGAT of $T_{init} = 1$s. Even spending only one extra second to optimise the solution is worse because all agents are delayed for 1s and therefore, the waiting cost adds up. Things get worse the longer we wait indicating that a minimal initial planning time (1s) is best.

**Experiment 3: How Long to Commit.** Committing to more steps of action may reduce flexibility in path improvement. This is because committed actions are locked, suggesting that always committing one step of action appears to be the optimal choice. Figure 3 shows the resulting relative SGAT with commit windows 1, 3, 10, 20. This metric is obtained by dividing the SGAT by the lower bound (the sum of single-agent shortest paths, disregarding other agents' paths).

Longer commit windows increase the number of LNS iterations per unit time, since we have relatively more time for improvement. We see that with a small number of agents, or where the path lengths of agents are short (Random), longer commit times are strictly worse; but when the number of agents is large and path lengths long, longer commit times are preferable. For example, we observe that for 1000 agents, in the first 20s, LNS is only able to run on average 10.5 iterations per second for Game, 5.7 for Warehouse and 29.2 for City, meaning in 1s it may be unable to complete even one iteration to improve the solution. Note that the number of iterations per second improves as the execution proceeds as agents remaining paths are shorter, and some agents are finished. For City, the path length is much longer than other maps, meaning the longer commits are even more preferable. This makes sense since missed opportunities to improve early commit windows that arise with long commit windows are ameliorated by getting more LNS iterations when paths to improve are longer.

Random

| $m$ | | 100 | 200 | 300 | 400 | 500 | 600 | 700 |
|---|---|---|---|---|---|---|---|---|
| AL | DU | 4.2 | 6.07 | 5.52 | 5.06 | 4.63 | 4.43 | 4.05* |
| AL | DI | 4.28 | 7.64 | 9.1 | 6.64 | **5.64** | **4.79** | **4.17** |
| AF | DU | 4.29 | 7.66 | 8.95 | 1.69 | 0.06 | 0.04 | 0.02 |
| AF | DI | **4.3** | **7.97** | **10.66** | **11.2** | 1.87 | 0.04 | 0.01 |

Warehouse

| $m$ | | 200 | 400 | 600 | 800 | 1000 | 1200 | 1400 |
|---|---|---|---|---|---|---|---|---|
| AL | DU | 2.04 | 3.14 | 3.93 | 4.44 | 4.77 | 4.48* | 4.07* |
| AL | DI | 2.33 | 4.65 | 5.93 | 4.84 | 5.24 | 4.98 | **4.42** |
| AF | DU | 2.33 | 4.72 | 6.66 | 7.77 | 5.92 | 0.41 | 0.44 |
| AF | DI | **2.34** | **4.75** | **6.82** | **8.89** | **11** | **12.4** | 0.13 |

Game

| $m$ | | 200 | 400 | 600 | 800 | 1000 | 1200 | 1400 |
|---|---|---|---|---|---|---|---|---|
| AL | DU | 1.77 | 2.55 | 3.09 | 3.24 | 3.62 | 3.78* | 3.88* |
| AL | DI | 1.84 | 2.86 | 3.12 | 3.3 | 3.49 | 3.75 | 3.8 |
| AF | DU | 1.85 | 3.57 | 4.58 | 2.54 | 0.37 | 3.78* | 3.88* |
| AF | DI | **1.86** | **3.64** | **5** | **5.32** | **5.18** | **4.1** | **3.89** |

City

| $m$ | | 500 | 1000 | 1500 | 2000 | 2500 | 3000 | 3500 |
|---|---|---|---|---|---|---|---|---|
| AL | DU | 2.36 | 4.62* | 6.87* | 8.88* | 10.91* | **12.24*** | **13.44*** |
| AL | DI | 2.39 | 4.66 | 6.92 | 8.91 | 10.76 | **12.24*** | **13.44*** |
| AF | DU | 2.41 | 4.59 | 6.87* | 8.88* | 10.91* | **12.24*** | **13.44*** |
| AF | DI | **2.41** | **4.78** | **7.21** | **9.53** | **12** | **12.24*** | **13.44*** |

Table 1: Throughput of differing Replanning and After-Goal strategies: **AL**: Replan All, **AF**: Replan Affected, **DU**: Stay at Dummy Goal, and **DI**: Disappear (both approaches are identical for $k = 1$). $m$ is the number of agents. "*" means the initial solver failed every commit, i.e. the solution is pure LaCAM$^\star$ partial solution and with no LNS improvement.

## Lifelong MAPF

For Lifelong MAPF, we do experiments on the same maps as one-shot MAPF. For each map, we extend the number of agents to test harder cases, including from 100 to 700, increased by 100 for Random, from 200 to 1400, increased by 200 for Warehouse and Game, from 500 to 3500, increased by 500 for City. For each map and number of agents, we generate problem instances by randomly selecting the same number of unique locations on the map as the number of agents as agent start locations. For the goal locations, we randomly select locations (which do not need to be unique) on the map. We fix the sequence of goals assigned to each agent for fair comparison. We set each move to be 1 second, and the simulation time to 1000 seconds. During the simulation, we only reveal one goal every time the agent finishes its current goal.

**Experiment 4: (Re-)Planning when Reaching a Goal.** We set commit = 1 as a baseline, and evaluate the throughput of Lifelong PIE with combinations of Replanning and After-Goal Decision strategies in Table 1. Note that for commit = 1 both Disappear strategies are identical. For the replan-

ner, Replan Affect outperforms Replan All in most cases. However, when adding more agents, e.g. for 500 agents in Random, Replan Affect finds it harder to replan paths and fix collisions. In such cases, Replan All maintains a stable improvement when scaling up. For the After-Goal Decision, Disappear is always better than Stay at Dummy Goal. This occurs because, under the Dummy Goal setting, low-level searches face challenges in finding paths, as the solution that traverses the goal then dummy goal requires a longer path to be computed, and collision avoidance has to consider target conflicts on dummy goals. Replan Affect and Disappear, which trades off maintaining the previous solution and more runtime for low-level searches, works the best in most cases. Replan All and Disappear become the best as we scale up. Note that for some cases when adding more agents, the initial solver always fails, and the solution is always the LaCAM⋆ partial solution. In these cases, we need to consider increasing the commit window to have more improvement when scaling up.

**Experiment 5: Throughput Evaluation.** Finally we compare lifelong PIE with RHCR, the existing start-of-the-art Lifelong MAPF solver, as well as an approach of simply replanning when a goal is achieved using LaCAM⋆ (effectively PIE with Replan All, and no path improvement). We consider two variants of PIE: one uses Replan Affected and usually generates the highest throughput ($PIE_F$); while the other uses Replan All and scales better eventually ($PIE_L$); details are given in Table 1. For RHCR, we set the parameter to be the best they reported ($w = 5, h = 10$ and low-level planner used is Priority-Based Search (Ma et al. 2019)).

As indicated in Table 1, for Disappear, we found when the number of actions per commit is more than 1, Disappear After Commit Window always generates better results than Disappear Immediately. Furthermore, Replan Affect with a smaller commit window, which gets benefits from knowing the next goals as quickly as possible, is better. While for Replan All, having more time to improve the LaCAM⋆ replanned solution (larger commit window) is more important and achieves more throughput.

Figure 5 shows the throughput achieved by LaCAM⋆, RHCR and lifelong PIE for different maps and numbers of agents. Clearly RHCR performs very well for small numbers of agents, but fails to scale at all well. $PIE_F$ is the highest throughput approach until the number of agents reaches a point where Replan Affected can not find a solution for affected agents within the commit time, where $PIE_L$ takes over. As numbers continue to increase $PIE_L$ degrades to effectively be equivalent to applying no path improvement (LaCAM⋆). Clearly the number of agents where $PIE_F$ starts to fail is different for different maps, and for City we never reach this point.

## Conclusions and Future Work

In this paper, we generate an efficient approach to planning and improving while executing for One-Shot MAPF and Lifelong MAPF problems. To do so we combine LaCAM⋆ (Okumura 2023) to generate initial solutions very fast, with MAPF-LNS (Li et al. 2021a) to improve solutions. The resulting algorithm provides substantially improved sum of

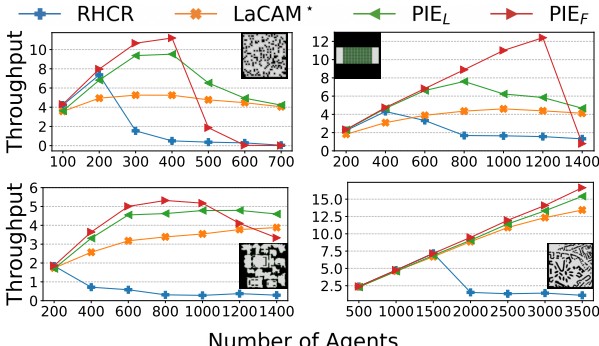

Figure 5: Throughput of RHCR, LaCAM⋆, $PIE_L$ and $PIE_F$.

|  | Random | Warehouse | Game | City |
|---|---|---|---|---|
| DA, All ($PIE_L$) | 10 | 20 | 20 | 20 |
| DA, Affect ($PIE_F$) | 1 | 1 | 1 | 3 |

Table 2: PIE Strategy with the associated best commits that achieved the throughput in Figure 5. "Best commits" refers to the commit window that has the most number of best throughput for different numbers of agents of one map. **DA**: Disappear After Commit Window. "All" and "Affect" means the replanning strategy. Column 1-4 is the associate commits that report the best performance. For commit = 1, Disappear After Commit Window is equivalent to Disappear Immediately.

Goal Achievement Times compared to optimising before executing for MAPF. We adapt the approach to Lifelong MAPF, by investigating how to replan agents when they reach their goal. We also make use of a Failure Resolution Policy to handle cases where a conflicting solution is found when committing to the next commit window. While replanning only affected agents is usually best, as the problem scales we simply have to throw away the previous solution, since we cannot replan new agents in the time available. Overall our Lifelong MAPF solution provides significantly greater throughput than competing approaches. Future work will extend PIE with dynamic commit windows and consider more planner speedups. In addition to this, considering execution with delay probabilities in PIE is an interesting direction for real-life applications.

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
