# OpenReview forum: "Planning and Execution in Multi-Agent Path Finding: Models and Algorithms"
_icaps-conference.org/ICAPS/2024/Conference — ICAPS 2024_

### Official Review · Reviewer_wTEv · 2024-01-17

**Significance And Importance:** 2
**Soundness:** 3
**Novelty:** 1
**Clarity:** 3
**Overall Evaluation:** 1
**Confidence:** 5

**Weaknesses:**

0: Minor weaknesses requiring some work to be addressed for the paper to be accepted.

**Contributions Of The Paper:**

This paper introduces a framework that considers both planning and execution time in the context of Multi-Agent Path Finding (MAPF) problems. The objective is to assess the effectiveness of a MAPF algorithm by incorporating the time needed to generate a plan, referred to as SAGT (which is commonly overlooked in literature), into the overall algorithmic runtime. This aspect is crucial for real-world applications. The framework assumes perfect plan execution by robots and permits replanning during execution. It introduces customizable parameters, such as the time allocated for replanning (i.e. commit window length), replanning strategy (i.e. replan all, replan affected) etc. Various experiments are conducted with different choices for these parameters.

The authors examine both the one-shot and Lifelong versions of MAPF, presenting a tailored approach for each that minimizes the SAGT objective. These approaches leverage existing MAPF algorithms, including LACAM*, MAPF-LNS variants, and PP.

The paper presents experimental results on four maps, showcasing the performance of their proposed schemes across different parameter choices. Additionally, comparisons are made with state-of-the-art methods from previous literature for the addressed problems.

**Ethical Considerations:**

(1) Not Applicable: The paper does not have any ethical considerations to address

**Nomination For Best Paper:**

No

**Questions For Authors:**

In experiment 5, when generating results with RHCR, is the RHCR algorithm being called during execution or after execution?

How frequently does MAPF LNS fail to find a feasible plan? My guess is that since the maps considered in the experiments have very few bottlenecks MAPF LNS would have a very high success rate. I would suggest including some discussions on this in the experimental results section. That way, the importance of using LACAM in line 7 of Algorithm 3 and the Failure resolution policy in line 15 can be understood from the solution quality perspective as well.

Experiment 1 suggests that a PIE scheme to generate an initial plan (before robots begin execution) can be better than applying MAPF-LNS in terms of SGAT. What can you say about using a PIE scheme to generate an initial plan (before robots start executing), and compare it with offline MAPF in terms of SOC?

Here are a few more issues I noted in the paper:
Pg 2
Line 86: Fix “iand”

Pg 4
Line 368-369: Simulator is first mentioned here, however it is not clear from the context what an execution simulator means.
Line 315: After-Goal Decision : This seems like an assumption that is not necessary in practice.
Line 320: In Lifelong, more than one agent may have the same goal location. 320 Applying MAPF planners to such problems will lead to search failure.
It is usually straightforward to adapt MAPF planners to handle the same goal.

Pg 5
Section on Failure Resolution Policy Based on MCP simply was very hard for me to understand despite reading it several times. I generally recommend writing this section more clearly.

Pg 6
Line 509: The sentence seems to have abruptly ended here.

**Reproducibility:**

4: Authors promise to release code and domains (whichever apply).

**Strengths Of The Paper:**

The paper presents convincing experimental results that suggest that when scaling to a large number of robots on maps with sparse bottlenecks, a concurrent planning and execution framework is very cost efficient when the cost is measured in terms of SAGT.

If I understand correctly, the main innovation in the approach presented in this paper compared to Rolling-Horizon Collision Resolution (RHCR) is that, for the portion of the plan beyond the temporal horizon there is a conscious effort to generate a conflict free plan whereas RHCR is only concerned about maintaining feasibility within the temporal horizon. This difference as the authors demonstrate can result in significant performance boost when measured in terms of SAGT.

**Weaknesses Of The Paper:**

The paper assumes an excessively optimistic scenario where robots flawlessly execute their plans, allowing all available execution time for replanning. However, in reality, robots may encounter failures, leading to a discrepancy between the anticipated and actual states after executing a committed section. Therefore, the practical utility of the PIE setup proposed in this paper becomes questionable.

The experimental results do not clearly elucidate the proposed scheme's dependence on the performance of MAPF-LNS. All experiments are confined to maps featuring sparse bottlenecks, which is an environment where MAPF-LNS excels. To comprehensively assess the algorithms' performance in congested settings, experiments on room and maze maps should be incorporated.

The section on the Failure Resolution Policy Based on MCP is not explained well in my opinion. I highly recommend writing this section more clearly.

---

> ### Author Rebuttal · Authors · 2024-01-28
>
> Thank you for your detailed evaluation of our paper. In response to your points:
> 1. Execution Failure:
> We acknowledge that it is an important feature of real MAPF applications. We have chosen not to work on execution failure, as developing algorithms for fully-featured MAPF problems is too complex in a single paper. We think PIE is a good starting point since we already have a policy for planning failures, and, in future work, we can extend this to handle execution failures.
>
> 2. Maps:
> We agree that room and maze create more bottlenecks. However, our experiments also include Warehouse, which has many corridors and intersections of width 1 and is an important real-world application for PIE. The random map is tested with up to 75% agents/free-space ratio, which is extremely dense. Due to the space limitation, we chose to show the performance in extreme situations on these maps with an increasing number of agents.
>
> 3. RHCR in Exp 5:
> The planning is after execution. We set the RHCR planning time limit to be the same amount as the execution time to simulate the simultaneous planning and execution process.
>
> 4. LNS2 fail frequency:
> Below is the success rate (s) of LNS2 in PIE$_F$ for different numbers of agents (n) from Exp 5. Note that LNS2 is only called when line 9 in Algorithm 3 fails. For City, the success rate is always 1.
>
> Random
>
> n& <500 & 500 & >500
>
> s& >0.97 & 0.08 & <0.01
>
>
> Warehouse
>
> n& <1000 & 1000 & 1200 & 1400
>
> s& >0.97 & 0.94 & 0.89 & <0.01
>
> Game
>
> n& <1000 & 1000 & 1200 & 1400
>
> s& >0.97 & 0.71 & 0.26 & 0.15
>
> We agree that it is important to explain LNS2 success rate to demonstrate the importance of LaCAM Replan and Failure Resolution Policy. We will add related discussions to the paper.
>
>
> 5. PIE as an initial planner:
> PIE benefits in terms of SGAT over LNS since it replans during execution.
> If we try to use PIE to generate an initial plan this advantage disappears, since there is no execution.
>
> 6. Line 315 and Line 320:
> In practice, multiple agents may share the same goal.
> Most MAPF solvers plan for agents to stay at goals forever. Thus, planning for agents with the same goal will cause errors, which need extra adaptation.
> Thus, from a planner's perspective, it is important to decide how to move the agent after reaching the goal without knowing the next goal, to allow planning feasible paths for other agents.
>
> 7. Presentation Issue:
> We will improve the writing of the failure policy and other suggested writing improvements for better clarity.

---

### Official Review · Reviewer_kcFV · 2024-01-22

**Significance And Importance:** 3
**Soundness:** 4
**Novelty:** 3
**Clarity:** 4
**Overall Evaluation:** 3
**Confidence:** 5

**Weaknesses:**

0: Minor weaknesses requiring some work to be addressed for the paper to be accepted.

**Contributions Of The Paper:**

The paper studies integrating planning and execution in MAPF and Lifelong MAPF.
The main contribution is the PIE framework, which is a real-time heuristic search framework having agents commit to a fixed number of steps
and plan for the next "window" during execution.
The authors then consider several design choices within PIE:
1. For planning the next steps to commit to, the authors integrate LaCAM* and LNS, using LaCAM* to find a solution fast and then LNS to improve it.
2. For choosing which agents should replan, the authors consider replanning for all agents and replanning only for a subset of agents
detected as affected by the upcoming planning period.
3. For deciding what to do after an agent reaches the goal, the authors consider assuming the agent disappeared or moving to a dummy goal.
4. For handling cases where the planner did not find a plan, the authors considered a novel MCP-based approach.

A very comprehensive set of experiments is performed on standard benchmarks to highlight the key take-home points:
1. Planning during execution is better (in terms of the "goal achievement time" metric) than offline planning, even for a little bit.
2. PIE allows a smooth anytime behavior, better than LaCAM* and MAPF-LNS2.
3. PIE yields significant advantages in terms of throughput as well.

Lastly, the authors present an intriguing failure policy based on MCP.

p.s. I love the link to the source code ;)

**Ethical Considerations:**

(1) Not Applicable: The paper does not have any ethical considerations to address

**Nomination For Best Paper:**

No

**Questions For Authors:**

1. Can you clarify the difference between PIE and RHCR?
2. Regarding the relation of PIE to Morag et al.'s work mentioned above
2.a. Can you discuss the difference between the Replan Affected strategy and their confllict-driven Agent Selection strategies?
2.b. Why didn't you compare the MCP-based failure policy and the simpler ones in Morag et al.'s work?
3. What can you say theoretically about your "Failure Resolution Policy Based on MCP" in terms of runtime and completeness?
4. I assume in a real application SGAT should be some combination of computation time and execution time, but giving different weights to them. What were these weights in the presented experiments?
I assume they were both 1, i.e., the authors sum the CPU time for planning and the CPU time for "executing". But if execution time was significantly slower than planning time, wouldn't that decrease the advantage of PIE? In particular, this would drastically the conclusions that can be drawn from Figure 2.

**Reproducibility:**

4: Authors promise to release code and domains (whichever apply).

**Strengths Of The Paper:**

I like this paper a lot and support accepting it.

1. It is very clearly written, and a joy to read.
2. The topic of planning during execution in MAPF is practical and under-studied.
3. The proposed planner in PIE yields very impressive results.
4. The experimental results are very impressive and the space of design choices explored and experiments gives nice insights to MAPF research.

**Weaknesses Of The Paper:**

I was very excited to read about the MCP-based failure policy, but it deserves a deeper analysis: what is its runtime? how effective is it?
Why is it not compared or evaluated to the simpler failure policies suggested by Morag et al. '23 ("Adapting to Planning Failures in Lifelong Multi-Agent Path Finding").

I am not fully convinced that PIE is different from RHCR.
PIE plans for the first k steps, then while moving it plans for the subsequent k steps.
RHCR plans the first k steps, executes them, and then plans for the next k steps.
Is this really different?

---

> ### Author Rebuttal · Authors · 2024-01-28
>
> We appreciate your thoughtful feedback on our paper. Regarding your points:
>
> 1. RHCR:
>
> Planning time:
> In the RHCR paper, the planning time is much longer than the execution, which means planning and execution are interleaved. In PIE, planning and execution times are equal to ensure concurrent planning. When setting them equal, RHCR can fail if the runtime limit is small, and RHCR does not handle planning failures. Also, RHCR stops on a feasible window plan, while PIE utilises all available time for continuous plan enhancement.
>
> Plan feasibility:
> RHCR solves a window problem, focusing on feasibility within the window. PIE ensures a feasible plan for each agent to reach their goal so that there is an overall improvement on the path.
>
> 2. Morag et al.'s work
>
> a:
> Replan Affect in PIE first replans agents with new goals. When it fails, it uses LNS2 to repair, including both collision agents and connected agents. Morag et al.'s Selection strategies select a fixed number of collision agents to replan, which does not include the connected agents.
>
> b:
> Morag et al.'s work only considers windowed plans, whereas we always have the full path for every agent, even if they are invalid. Their failure policy does not produce the full path that PIE is expecting, while the MCP-based policy keeps the full path by inserting delays. It is similar to IStay while we maintain the whole plan. AllStay is worse than approaches that allow some agents to move (IStay, IAvoid).
>
> 3. MCP:
> Our MCP implementation touches every agent once along every location on its final path, thus it is O(SOC) where SOC is the sum of costs from the current state. It runs very fast in practice (a few milliseconds). We tried limiting the MCP simulation to just the next time window, with complexity O(n*k), but we found this to be too myopic, as it led to many later conflicts to resolve.
>
> 4. Execution and computation time:
> In the PIE lifelong experiment, execution and computation time for a k-step window are identical, which means 1 timestep comes with 1 second for planning.
> If robots move slower this will increase computation time for us, as well as RHCR.
> Given the excellent performance of LNS in improving paths, this extra time will help PIE just as it does RHCR, but it is true that if both methods are given a large amount of computation time, the differences will be smaller.
> Note that PIE uses additional time to improve solutions, while RHCR will stop computation once it finds a solution for the window.

---

### Official Review · Reviewer_RkKF · 2024-01-23

**Significance And Importance:** 2
**Soundness:** 3
**Novelty:** 2
**Clarity:** 3
**Overall Evaluation:** 1
**Confidence:** 5

**Weaknesses:**

1: Minor weaknesses that are easily fixable.

**Contributions Of The Paper:**

The paper studies the problem of multi-agent pathfinding (MAPF), specifically, the real-world task of applying a found plan on real agents. The issue the paper raises is that in the previous studies, the computation time is not counted towards the total runtime of the task. Instead, they propose that the total runtime should be measured as both the computation time and the execution time. To this end, they propose an algorithm that tries to minimize this sum by starting the execution as soon as possible, even following an imperfect plan, improving the plan during the execution. The proposed algorithm makes use of off-the-shelf MAPF algorithms, specifically LaCAM* to quickly find an initial plan, and MAPF-LNS(2) to improve this plan during execution. They extend the algorithm to work both in offline settings as well as lifelong settings. Experiments comparing different parameters of the algorithm are also included.

**Ethical Considerations:**

(1) Not Applicable: The paper does not have any ethical considerations to address

**Nomination For Best Paper:**

No

**Questions For Authors:**

1. See my comment about SGAT in the weaknesses section.

*** After rebuttal
Thank you for clarifying points from the review

**Reproducibility:**

4: Authors promise to release code and domains (whichever apply).

**Strengths Of The Paper:**

- The paper raises an important real-world application question about the research in the area of MAPF. Indeed, there are settings where the computation of the plan plays a huge role, and should be considered as well as the execution time.
- A (quite simple) algorithm for the offline setting is provided.
- The authors also study the problem of lifelong planning, which is more challenging due to the agent receiving new tasks, possibly at inconvenient times. Non-trivial modifications to the offline algorithm were made.

**Weaknesses Of The Paper:**

- The paper provides no theoretical results about the proposed algorithms (Complete? If so, when? In which cases it is not?). Granted, these depend mostly on the underlying MAPF solvers, but it still should be stated if your algorithm changes any of these properties.
- As in the point above, the algorithm depends heavily on the used MAPF solvers, therefore, these should be clearly described for the paper to be self-contained. In the current version, the description could be improved. A reader who is not familiar with the algorithms will have a hard time understanding the rest of the paper.
- I question the definition of SGAT. The GAT is meant for a single agent and in that setting it makes sense. However, counting the time it takes to compute the initial solution as many times as there are agents seems too harsh. I understand that the motivation comes from the sum of costs, however, SOC is often linked to fuel consumption and it can be argued that the agents do not consume any fuel before the first actions are received. Perhaps makespan would be a better function to optimize? Note that for a single agent, SOC and makespan are identical, so moving from GAT to SGAT (in SOC manner) should be better justified.


Typos I spotted:
- line 28 "When solving MAPF problems existing studies" missing comma
- line 86 "iand"
- line 87 "thoughput"
- line 87 "MAPF, in comparison" extra comma
- line 126 "(PP) (Silver 2005) PP" missing period
- line 216 "Agents" should not be capitalized
- line 299 "Noticing" Notice
- line 398 "agents Our" missing period
- line 473 "order or v" of
- line 613 "start-of-the art" state

---

> ### Author Rebuttal · Authors · 2024-01-28
>
> We appreciate your positive feedback and valuable suggestions. Addressing your points:
> 1. SGAT:
> We agree that other objective functions, such as Makespan, may make more sense than SGAT for the one-shot MAPF, depending on the application.
> However, we do think that SGAT is a natural extension of the Sum of Costs to take into account both the delay before the start and the execution time.
> Furthermore, we want to point out that the ratio of computing time to travel time is about the same in SGAT and Makespan. Indeed, SGAT sums the computing time for each agent, but it also adds path lengths for all agents.
> We can also adjust our solver to optimise the Makespan, the danger of using Makespan as the objective is that the solver may only optimise the path of one agent which has the longest path length.
> In Lifelong MAPF, we use throughput to measure the performance.
>
> 2. Algorithms completeness:
> PIE relies on the fact that Lacam* is itself complete, and on the fact that LNS only changes a feasible solution to another feasible solution. Therefore, for One-shot MAPF, PIE is also complete.
>
> For Lifelong MAPF, in case we switch to Lacam* for replanning (replan all), then PIE is complete. But in case we use MAPF-LNS2 and don't switch to Lacam* (replan affect), then PIE is not complete, because MAPF-LNS2 is not complete (it uses prioritized planning as the repair solver).
> Subject to paper length, we will add a short discussion on completeness.
>
> 3. Description of MAPF Solvers:
> We understand the importance of clarity in describing the underlying MAPF solvers. We will enhance their descriptions to ensure better comprehension for readers unfamiliar with these algorithms.

---

### Meta-Review · Area_Chair_di4K · 2024-02-02

**Recommendation:** Accept (Oral)
**Confidence:** 1

**Metareview:**

The paper proposes a new way to look on interleaving planning and execution in MAPF.

Strengths: New direction. nicely Written. Significant experimental results.
Weaknesses: Some worries about the connection to RHCR and there is no theoretical analysis

As a whole, all the reviewers liked this paper and we recommend accepting it.

**Ethical Considerations:**

(1) Not Applicable: The paper does not have any ethical considerations to address